# Probing cardiomyocyte mobility with multi-phase cardiac diffusion tensor MRI

**Kévin Moulin**[1,2]*, **Ilya A. Verzhbinsky**[1,2], **Nyasha G. Maforo**[2,3], **Luigi E. Perotti**[4], **Daniel B. Ennis**[1]

**1** Department of Radiology, Stanford University, Stanford, CA, United States of America, **2** Department of Radiological Sciences, University of California, Los Angeles, CA, United States of America, **3** Physics and Biology in Medicine IDP, University of California, Los Angeles, CA, United States of America, **4** Department of Mechanical and Aerospace Engineering, University of Central Florida, Orlando, FL, United States of America

* kevin.moulin.26@gmail.com

**Data Availability Statement:** Raw and reconstructed dataset used in this study can be found at https://med.stanford.edu/cmrgroup/data.html.

## Abstract

### Purpose

Cardiomyocyte organization and performance underlie cardiac function, but the *in vivo* mobility of these cells during contraction and filling remains difficult to probe. Herein, a novel trigger delay (TD) scout sequence was used to acquire high in-plane resolution (1.6 mm) Spin-Echo (SE) cardiac diffusion tensor imaging (cDTI) at three distinct cardiac phases. The objective was to characterize cardiomyocyte organization and mobility throughout the cardiac cycle in healthy volunteers.

### Materials and methods

Nine healthy volunteers were imaged with cDTI at three distinct cardiac phases (early systole, late systole, and diastasis). The sequence used a free-breathing Spin-Echo (SE) cDTI protocol (b-values = 350s/mm$^2$, twelve diffusion encoding directions, eight repetitions) to acquire high-resolution images (1.6x1.6x8mm$^3$) at 3T in ~7 minutes/cardiac phase. Helix Angle (HA), Helix Angle Range (HAR), E2 angle (E2A), Transverse Angle (TA), Mean Diffusivity (MD), diffusion tensor eigenvalues ($\lambda_{1-2-3}$), and Fractional Anisotropy (FA) in the left ventricle (LV) were characterized.

### Results

Images from the patient-specific TD scout sequence demonstrated that SE cDTI acquisition was possible at early systole, late systole, and diastasis in 78%, 100% and 67% of the cases, respectively. At the mid-ventricular level, mobility (reported as median [IQR]) was observed in HAR between early systole and late systole (76.9 [72.6, 80.5]° vs 96.6 [85.9, 100.3]°, p<0.001). E2A also changed significantly between early systole, late systole, and diastasis (27.7 [20.8, 35.1]° vs 45.2 [42.1, 49]° vs 20.7 [16.6, 26.4]°, p<0.001).

### Conclusion

We demonstrate that it is possible to probe cardiomyocyte mobility using multi-phase and high resolution cDTI. In healthy volunteers, aggregate cardiomyocytes re-orient themselves

**Funding:** This work was supported by NIH/NHLBI R01-HL131975, R01-HL131823, K25-HL135408 grants, by NSF DGE 1650604 grant, and by AHA 20POST35210644 grant. The content of this manuscript is solely the responsibility of the authors and does not necessarily represent the official views of the National Institutes of Health.

**Competing interests:** The authors have declared that no competing interests exist.

more longitudinally during contraction, while cardiomyocyte sheetlets tilt radially during wall thickening. These observations provide new insights into the three-dimensional mobility of myocardial microstructure during systolic contraction.

## Introduction

Myocardial microstructure is complex and highly organized, composed of a continuously branching and merging syncytium of cardiomyocytes that are organized in layers called "sheetlets". The predominant long-axis orientation of aggregate cardiomyocytes, also referred to as the "myofiber" direction, changes transmurally from epicardium to endocardium [1]. In the left ventricle (LV), the myofiber orientation can be characterized by measuring the helix angle (HA) and the transverse angle (TA). The HA represents the orientation of the myofiber (projected in the local tangent plane) with respect to the circumferential direction. The TA measures the angle between the myofiber direction projected onto the local horizontal plane (normal to the epicardium) and the circumferential direction [2]. The laminar sheetlet structure of the heart is usually represented by the angle formed between the sheetlet orientation and the cross-myocyte direction [3].

Cardiomyocyte organization and performance underlie cardiac function, but the *in vivo* mobility (i.e., change in orientation) of these cells during contraction and filling remains difficult to probe and poorly described. Histological studies have described a diverse range of tissue-level phenotypes in diseased hearts [4–6], which may explain the equally diverse array of abnormal *in vivo* deformations reported in the failing LV [7]. Examining *in vivo* LV microstructural kinematics in the beating heart may help to differentiate purported tissue-level mechanisms of cardiac function and dysfunction in healthy and diseased hearts [8].

Magnetic resonance diffusion-weighted imaging (DWI) has been recently adapted for cardiac applications [9], where it continues to emerge as a technique to investigate *in vivo* myocardial microstructure and remodeling. In particular, cardiac diffusion tensor imaging (cDTI) has become an insightful method for determining cardiomyocyte orientations in healthy [10] and pathologic [3] hearts. Using cDTI, the impaired mobility of aggregate cardiomyocytes during contraction has been observed in patients with suspected hypertrophic cardiomyopathy, dilated cardiomyopathy [8], and amyloidosis [11], suggesting a biomechanical link between cardiomyocyte mobility and LV dysfunction.

Recent advances make it possible to acquire reliable cDTI *in vivo*, but cDTI at high spatial resolution (<2mm isotropic in-plane) remains challenging. Two distinct techniques have emerged to acquire cDTI *in vivo* and minimize the image signal loss due to cardiac motion. The STimulated Echo Acquisition Mode (STEAM) method distributes the encoding/decoding over two consecutive heartbeats to minimize the effect of bulk cardiac motion. The STEAM approach is challenged by lower SNR, longer acquisition times and challenges that limit free-breathing approaches. The very short diffusion encoding gradients, however, enable imaging at almost any phase of the cardiac cycle. More recently, higher spatial resolution has been achieved with new sampling strategies like multi-shot spiral imaging [12].

Alternatively, Spin Echo (SE) DWI approaches [13–15] using a first and second-order motion compensated gradient design (M1 = M2 = 0) have also been applied to *in vivo* cDTI. The SE approach produces higher SNR than its STEAM counterpart [9, 16] and can be adapted for free-breathing, both of which make it more amenable to higher spatial resolution acquisitions. However, two limitations restrict the use of the M1-M2 compensated SE approach. First, the M1-M2 compensated gradient design incurs a TE penalty, which requires optimizing both

the sequence parameters and the gradient design [13, 14]. The second limitation inherent to M1-M2 compensated gradient waveforms is that it is assumed that the underlying cardiac motion is comprised of only constant velocity or acceleration during encoding. Consequently, M1-M2 compensated gradients lead to intravoxel signal loss when cardiac motion is more complex. In general, these assumptions may only be true during a subset of cardiac phases, which differ from patient to patient or even beat to beat. This subject-specific dependence results in 'on-the-fly' manual adjustments to find the correct trigger delay (TD) for high quality imaging, adding to the overall acquisition time and increasing protocol complexity.

The objective of this work was to acquire SE high-resolution cDTI data at multiple cardiac phases to characterize cardiomyocyte organization and mobility at several cardiac phases. A prospective TD scout approach was used to estimate the patient specific cardiac phases available for cDTI. Two to three distinct cardiac phases were acquired in healthy volunteers in which *in vivo* cardiomyocyte mobility during contraction and filling was characterized.

## Methods

### Imaging protocol

Nine healthy volunteers (N = 9) were recruited for an MRI exam after obtaining signed statements of informed consent under an IRB and HIPPA compliant protocol approved by the University of California Los Angeles (UCLA) ethic committee (IRB#14–001743). All acquisitions were performed on a single 3T MRI scanner (Prisma, Siemens), using an 18-channel body phased-array coil and a 32-channel spine array coil. Long-axis, four-chamber, and short-axis localizers, as well as short-axis balanced steady-state free precession (bSSFP) cine images were first acquired under breath-hold conditions. A short-axis cine DENSE sequence (resolution 2.3 x 2.3 mm, 3D four-point displacement encoding, $k_e = 0.8$, TE = 1.05ms, TR = 15ms, 30 cardiac phases or more) was also performed using free-breathing and an expiratory triggered navigator for a subset of subjects (N = 5). The measured displacements were used to estimate the longitudinal cardiac displacement between the cardiac phases of interest.

### cDTI sequence

The cDTI acquisition parameters were optimized to minimize signal dropout induced by cardiac motion and achieve higher spatial resolution. A custom single-shot spin-echo EPI sequence was modified to incorporate second-order ($M_1$-$M_2$) motion compensated diffusion encoding gradients [14, 15], inner-volume reduced-field-of-view (rFOV) [17], prospective navigator echo for respiratory triggering (acceptance windows 4 mm during expiration) and fat saturation. First, the sequence parameters were chosen so that the combination of the rFOV, the readout bandwidth, partial Fourier factor, and GRAPPA [18] parallel imaging minimized the EPI echo train length. The resultant short EPI echo train length limited the image distortion caused by high resolution and motion blurring. This approach provides a minimal TE contribution from the readout (Fig 1C), which permits a longer time for diffusion encoding, and reduces both the overall TE and diffusion encoding temporal footprint of the sequence. Second, the EPI readout distortion was further reduced by using a cardiac $B_0$ shim focused on the left ventricle. Lastly, the $M_1$-$M_2$ gradient waveform was optimized on-the-fly symmetrically to reach a second gradient moment ($M_2$) as close to zero as possible as defined by Welsh *et al.* [13].

For each subject, one mid-ventricular short-axis slice was imaged at different cardiac phases. The acquisition parameters were: TE = 61 ms, rFOV 200x160 mm$^2$, acquisition matrix 128x104, resolution 1.6x1.6x8 mm$^3$ (interpolated in-plane to 0.8x0.8x8 mm$^3$), parallel imaging with 2x-GRAPPA, partial Fourier 6/8, and readout bandwidth of 1860 Hz/px. Two b-values (0

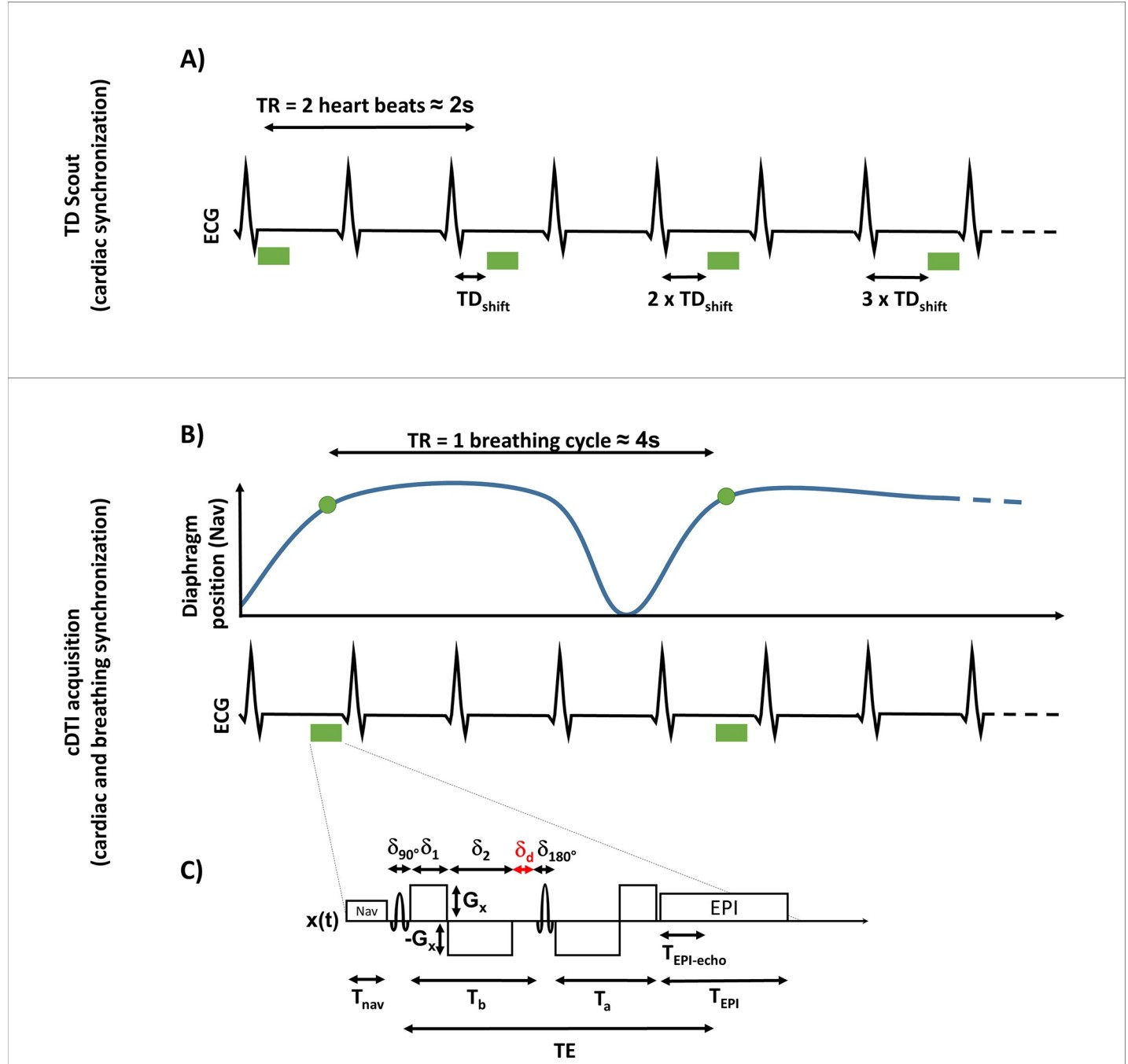

**Fig 1.** (A) The TD scout was acquired during free-breathing with cardiac synchronization. Images were acquired every two heartbeats to allow for signal recovery (TR≈2s). (B) The cDTI acquisitions was cardiac and respiratory triggered. One image was acquired every respiratory cycle (TR≈4s). (C) Details of the cDTI sequence used in this work. The timings of the EPI acquisition $T_{nav}/\delta_{90°}/\delta_{180°}/T_a/T_b/T_{EPI\text{-}echo}/T_{EPI}/T_E$ were 8/2/4/24/20/9/23/61ms and the timings of the repeated diffusion encoding gradient $\delta_1/\delta_2$ were 7.2/12.6ms. The deadtime $\delta_d$ was 4ms.

and 350 s/mm$^2$) were acquired using twelve diffusion encoding directions and eight repetitions. The diffusion encoding duration with motion compensation was 48 ms while the EPI readout was 23 ms. After optimization, the four diffusion encoding gradients have total durations of 7.2/12.6/12.6/7.2 ms with a ramp time of 1.52 ms, corresponding to gradient moment

$M_0$ = 0 mT/m/s, $M_1$ = 1.3 mT/m/s$^2$, $M_2$ = 62.6 mT/m/s$^3$ with an amplitude G = 71.3 mT/m (for reference a monopolar gradient waveform with the same sequence parameters has $M_0$ = 0 mT/m/s, $M_1$ = 11.5 x 10$^3$ mT/m/s$^2$, $M_2$ = 555.9 x 10$^3$ mT/m/s$^3$ and G = 22.5 mT/m). The single-shot acquisition was ECG and end-respiratory triggered (Fig 1B) using a navigator located on the right liver dome resulting in a TR of one breathing cycle (~4000 ms). A total of 104 images were obtained per cardiac phase corresponding to an acquisition time of ~7 minutes.

## TD scout acquisitions

To precisely determine which cardiac phases were suited for cardiac diffusion imaging, a prospective TD scout acquisition was implemented [15, 19]. As shown in Fig 1A, the TD scout consists of repeated diffusion acquisitions where the TD between the electrocardiogram R-wave and the excitation pulse of the diffusion sequence is sequentially shifted at each acquisition. The TD scout was acquired during free-breathing without navigator triggering and one image was acquired every two heartbeats (TR ≈ 2000ms for a heart rate of 60 bpm). A total of thirty cardiac phases were scouted over the whole cardiac cycle by adjusting the TD shift to the heartbeat. For each cardiac phase, three orthogonal diffusion directions at b-value 350 s/mm$^2$ were acquired. The other sequence parameters were kept the same as the ones used for the cDTI acquisition described above. The total scan time for the TD scout was ~3 minutes for the corresponding 30 cardiac phases and 90 images.

In this work, three cardiac phases were studied: 1) early systole after the R-wave (TD~150–200 ms); 2) late systole as close as possible to maximum contraction (TD~250–350 ms); and 3) diastasis (TD~700–900 ms). Within these ranges, the TD for each cardiac phase was determined by visual assessment of the Trace signal (average across diffusion directions) of the TD scout acquisition. The TDs were identified by selecting the TD scout images with the least motion induced artifacts. Imaging all three cardiac phases in every volunteer was not always possible due to subject specific cardiac motion. However, at least two cardiac phases were successfully acquired in each volunteer.

## Post-processing

All analyses and cDTI reconstruction were performed off-line using custom software (Matlab, Mathworks, repository: https://github.com/KMoulin/DiffusionRecon). Raw and reconstructed dataset can be found at https://med.stanford.edu/cmrgroup/data.html.

For the retrospective quantitative analysis of the TD scout, the mean signal over the central zone of the DWI image was reported (see S1 Fig). Subsequently, the reported mean signals were normalized with respect to the maximum mean signal across all cardiac phases per volunteer.

For the cDTI acquisition, image post-processing included rigid registration, repetition averaging, and an in-plane interpolation using Fourier transform zero-filling. The twelve resulting diffusion encoded images and the non-diffusion weighted image were then fitted to the tensor model using a non-linear least-squares method [20]. The mean diffusivity (MD), the fractional anisotropy (FA), the eigenvectors ($E_1$, $E_2$, $E_3$), and the eigenvalues ($\lambda_1$, $\lambda_2$, $\lambda_3$) were extracted from the tensor model. The first eigenvector $E_1$ of the cDTI model represents the myofiber orientation, while the second eigenvector $E_2$ is usually attributed to the sheetlet direction. $\lambda_1$, $\lambda_2$, and $\lambda_3$ represent, the diffusivity along primary, secondary, and tertiary diffusion directions respectively.

Endocardial and epicardial LV regions of interest (ROIs) were manually drawn to exclude the blood pool, the papillary muscles, and the right ventricle (RV) (Fig 2A). The heart was divided into six segments defined by the American Heart Association (AHA) guidelines [21]. Voxel-wise helix angle (HA), $E_2$ angle (E2A), and transverse angle (TA) (Fig 2B) were computed after projecting $E_1$ onto the epicardial tangent (HA) and the horizontal plane normal to

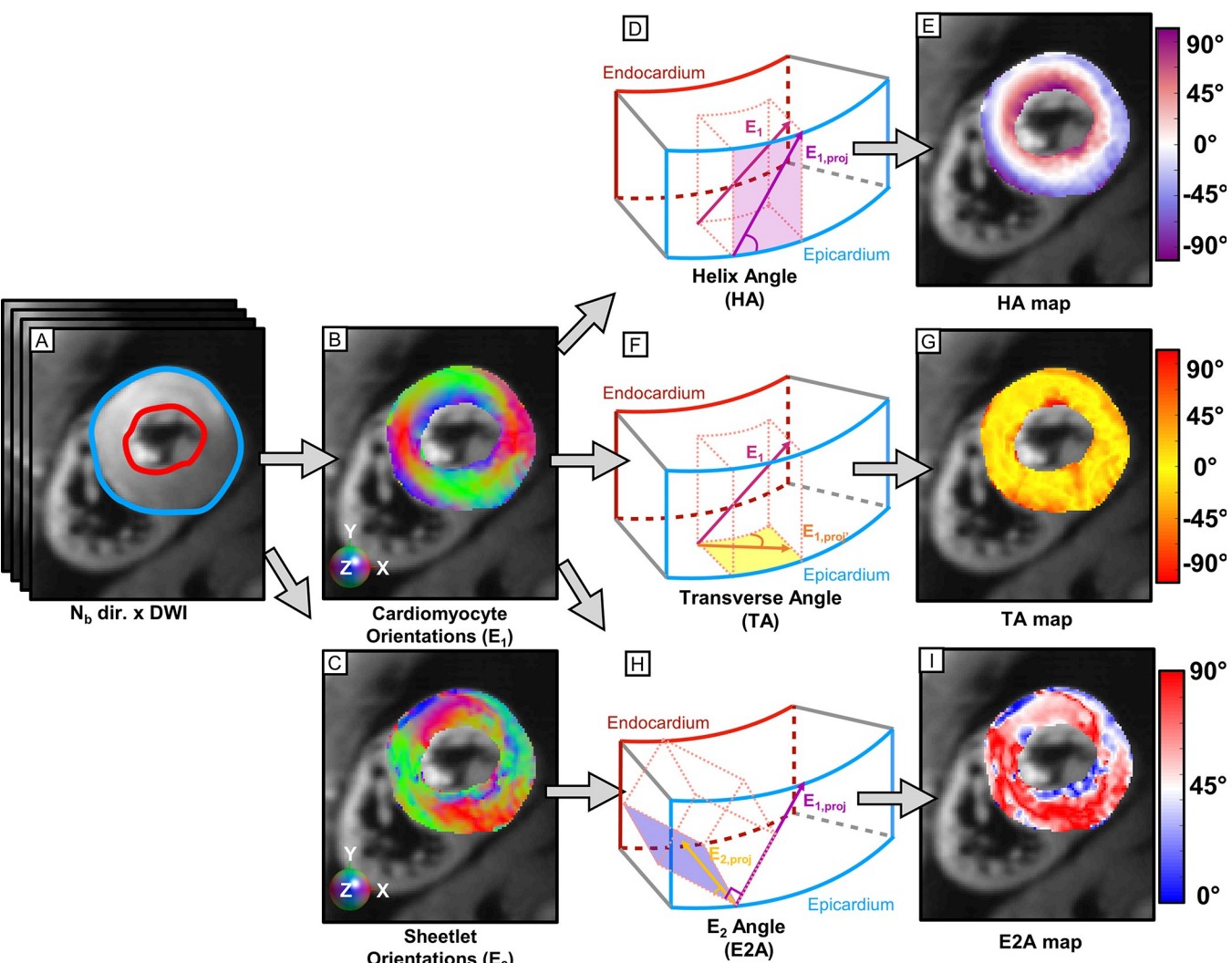

**Fig 2. Calculation of HA, TA, and E2A at each cDTI voxel.** (A) Stack of $N_b$ diffusion weighted images with an example LV ROI. The epicardial surface is shown in blue and the endocardial surface in red. (B-C) RGB colormap of the primary and secondary eigenvector orientation from the reconstructed diffusion tensors in the mid LV SA slice. (D-F-H) HA, TA, and E2A were calculated at each voxel after projection of the primary and secondary eigenvectors ($E_1$ and $E_2$) [2, 3, 22]. (E-G-I) HA, TA, and E2A maps in a representative volunteer (E2A is reported as absolute value).

the epicardium (TA), and after projecting $E_2$ onto the plane normal to $E_1$ projection (E2A) [2, 3, 22] (Fig 2D). Lastly, a filter was applied to unwrap the HA (S2 Fig).

## Quantitative analysis

Transmural HA distributions in the mid-ventricular slice for each subject were reported as a function of the normalized transmural wall depth measured from the epicardial and endocardial contours [22]. The HA distribution was described using nine points equally spaced along the transmural wall. For each point, the median HA was reported. For simplicity, only the epicardial, mid-myocardial, and endocardial medians are reported in the text and tables.

The HA distributions were also characterized by the HA range (HAR), which describes the angle difference between the endocardium and epicardium. HAR was calculated for each volunteer by subtracting the median HA at the epicardium from the median HA at the endocardium. Between two cardiac phases, the myofiber mobility was then calculated as the relative

change in HAR (ΔHAR). ΔHAR and HAR were computed for each volunteer and then reported across volunteers.

Finally, all quantitative values (ΔHAR, HAR, E2A, TA, MD, FA, $\lambda_1$, $\lambda_2$, and $\lambda_3$) are reported as median [Quartile 1, Quartile 3] across all volunteers for the three cardiac phases. All statistical analyses were carried out in SPSS (Version 26, IBM Corp.) using a non-parametric Kruskal-Wallis one-way ANOVA. For statistically different distributions, corresponding pair-wise comparisons were realized between cardiac phases using a Bonferroni post hoc correction. $p < 0.05$ was considered statistically significant.

### Reproducibility of helix angle and E2A

An intra- and inter-observer study was performed to estimate the reproducibility of HAR and E2A. Two manual segmentations (intra-observer) were performed by two observers (inter-observer) for a total of four observations per subject. Median HAR, E2A as well as intra-class correlation coefficient (ICC) were reported across subjects for the four observations. The first segmentation from the first observer was used for all non-reproducibility analyses reported in the results section.

## Results

### TD scout and cDTI acquisitions

A total of nine healthy volunteers were scanned (4 males and 5 females, 28 ± 4 years old, 60 ± 11 beats per minute). First, the subject specific TDs were determined by using the TD scout sequence. Because they were during free-breathing (without a navigator) with less recovery time (TR ~two heart beats instead of four), the TD scout images had lower signal and more non-motion related artifacts than the subsequent cDTI images. However, it was possible to visually identify motion related artifacts from the TD scout images. An example of the TD scout acquisition and the corresponding average signal per TD are displayed in Fig 3. As shown in Fig 3A and 3B, cardiac motion signal dropout was subject dependent and thus all three cardiac phases were not available for every volunteer. After visual assessment of the TD scout image quality, it was determined prospectively that cDTI acquisition was possible at late systole, early systole, and diastasis in 100%, 78% (7 out of 9 volunteers), and 67% (6 out of 9 volunteers) of the cases, respectively. The corresponding TDs across all volunteers for early systole, late systole and diastasis were 150 [148, 153] ms, 318 [293, 329] ms and 814 [800, 855] ms respectively. cDTI was more reliably acquired at late systole, while early systole and diastasis presented more motion artifacts (Fig 3C). Four volunteers had high quality data during all three cardiac phases, three at early and late systole, and finally two at late systole and diastasis. The mean acquisition time for the high-resolution cDTI protocol per cardiac phase was 7 ± 2 min. Representative cardiomyocyte orientations $E_1$, HA, E2A, and TA maps for one volunteer at all three cardiac phases are presented in Fig 4.

Due to longitudinal cardiac motion, the three imaged cardiac phases may have been acquired at slightly different positions along the LV long axis. Using the DENSE acquisition from a subset of the volunteers (N = 5), the mid-ventricular longitudinal motion was estimated to be 4 mm from early to late systole, 4 mm from diastasis to early systole, and -8 mm between late systole and diastasis (positive values correspond to motion from base towards apex).

### Helix angle and helix angle range

The HA distributions as a function of wall depth were extracted from the HA maps. Fig 5 shows an example HA distribution for the six AHA segments and for the entire mid-

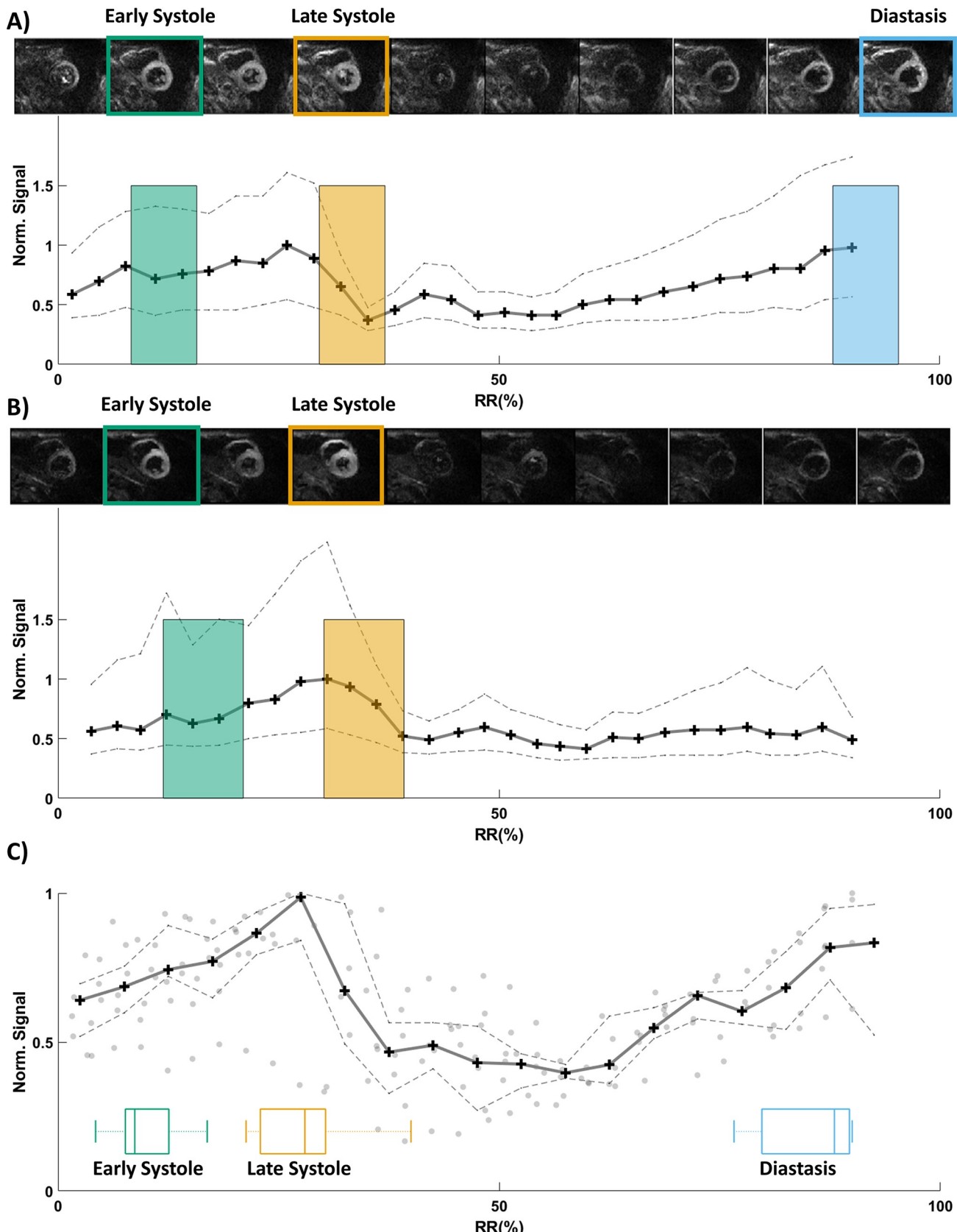

**Fig 3. Examples of TD scout acquisition and corresponding signal as a function of TD normalized over the RR interval.** Representative cases in which it was possible (A) and not possible (B) to acquire cDTI during diastasis. The signal is reported as the median DWI signal (solid line) and interquartile range [Q1 Q3] (dashed line). The signal is computed across all images per volunteer after segmentation (see S1 Fig) and after normalization by the cardiac phase with the mean highest signal. The colored boxes represent the temporal windows in which the cDTI has been acquired. Thus, the width of each box corresponds to the time from triggering to TE normalized with respect to the RR interval. Panel (C) shows the overall signal measured across all volunteers as a function of TD normalized over the RR interval. Median (solid line) and [Q1 Q3] range (dashed line) TD scout signals measured across all volunteers as well as individual values (points) corresponding to each volunteer. Boxplots show the distribution of TDs for early systole, late systole, and diastasis across all volunteers.

ventricular myocardium. A smaller IQR was observed in each AHA segment compared to the global HA distribution, demonstrating that the transmural HA distributions are segment specific. Additionally, the shape of the HA distribution for each AHA segment is largely conserved across each imaged cardiac phase as, for example, in the anterolateral segment (Fig 5). The epicardial myofibers in the anterior and inferior segments display a wide distribution due, in part, to the right ventricle (RV) insertions (Figs 5 and 6). Table 1 reports HA and HAR measured across volunteers in the six AHA segments. The intra-segment HAR was maximum at late systole compared to early systole and diastasis for the anterior, inferior, and anterolateral segments. In contrast, the intra-segment HAR was maximum in diastasis compared to early and late systole for the inferolateral and inferoseptal segments. The anterior segment had the overall highest HAR at late systole 111.3 [91.9, 119]˚. The inferior segment presented the lowest HAR in the three cardiac phases (47.4 [40.7, 54.0]˚ vs 87.5 [66.4, 96.0]˚ vs 67 [63.3, 82.7]˚) for early systole, late systole, and diastasis, respectively.

Regarding myofiber mobility between cardiac phases as shown in Table 2, ΔHAR was greatest between diastasis and early systole in the inferoseptal segment -37.4 [-69.5, -7.4]˚. The anteroseptal wall was the least mobile segment with ΔHAR equal to -1.1 [-9.6, 17.3]˚, 9.7 [-13.0, 9.0]˚, and -12.1 [-19.2, -4.1]˚ from early systole to late systole, from late systole to diastasis, and from diastasis to early systole, respectively.

Considering the entire mid-ventricular myocardium, the average endocardial, mid-wall, and epicardial HA across all volunteers was (44.4 [39.2, 45.9˚]; 2.1 [-2.1, 7.3]˚; -33.8 [-34.7, -32.0]˚) in early systole, (53.3 [45.8, 59.5]˚; 5.9 [3.7, 8.4]˚; -41.3 [-50.7, -39.7]˚) in late systole, and (55.0 [51.2, 55.4]˚; 8.8 [1.4, 9.4]˚; -33.8 [-34.7, -32.0]˚) in diastasis. HAR was found to increase from early systole to late systole (76.9 [72.6, 80.5]˚ vs 96.6 [85.9, 100.3]˚), and subsequently slightly decreased from late systole to diastasis (91.7 [85.9, 100.8]˚) as shown in Fig 7A. ΔHAR between early and late systole was larger in magnitude (21.8 [17.1, 25.4]˚) than between late systole and diastasis (-4.2 [-1.3, 3.7]˚) or between diastasis and early systole (-11.8 [-29.0, 3.5]˚) as seen in Table 2. Overall, statistical differences were found in HAR across cardiac phases (p = 0.004). Pair-wise comparisons were significant between early and late systole (p = 0.005) and between diastasis and early systole (p = 0.004), but no statistical difference was detected between late systole and diastasis (p = 1).

## Transverse angle and E2 angle

Fig 7B–7C shows the inter-volunteer distribution of TA and E2A at early systole, late systole, and diastasis. Similar TA values were observed across cardiac phases (-3.8 [-9.0, -2.2]˚ at early systole, -1.5 [-2.1, -0.1]˚ at late systole, and -0.8 [-1.3, 0.4]˚ at diastasis). Overall, TA distributions were statistically different among cardiac phases (p = 0.033), but no pair-wise differences were observed after post-hoc correction between early and late systole (p = 0.094), late systole and diastasis (p = 1), and diastasis and early systole (p = 0.054).

In contrast, an increase was observed in E2A distribution from early systole to late systole (27.7 [20.8, 35.1]˚ vs 45.2 [42.1, 49.0]˚) followed by a decrease from late systole to diastasis

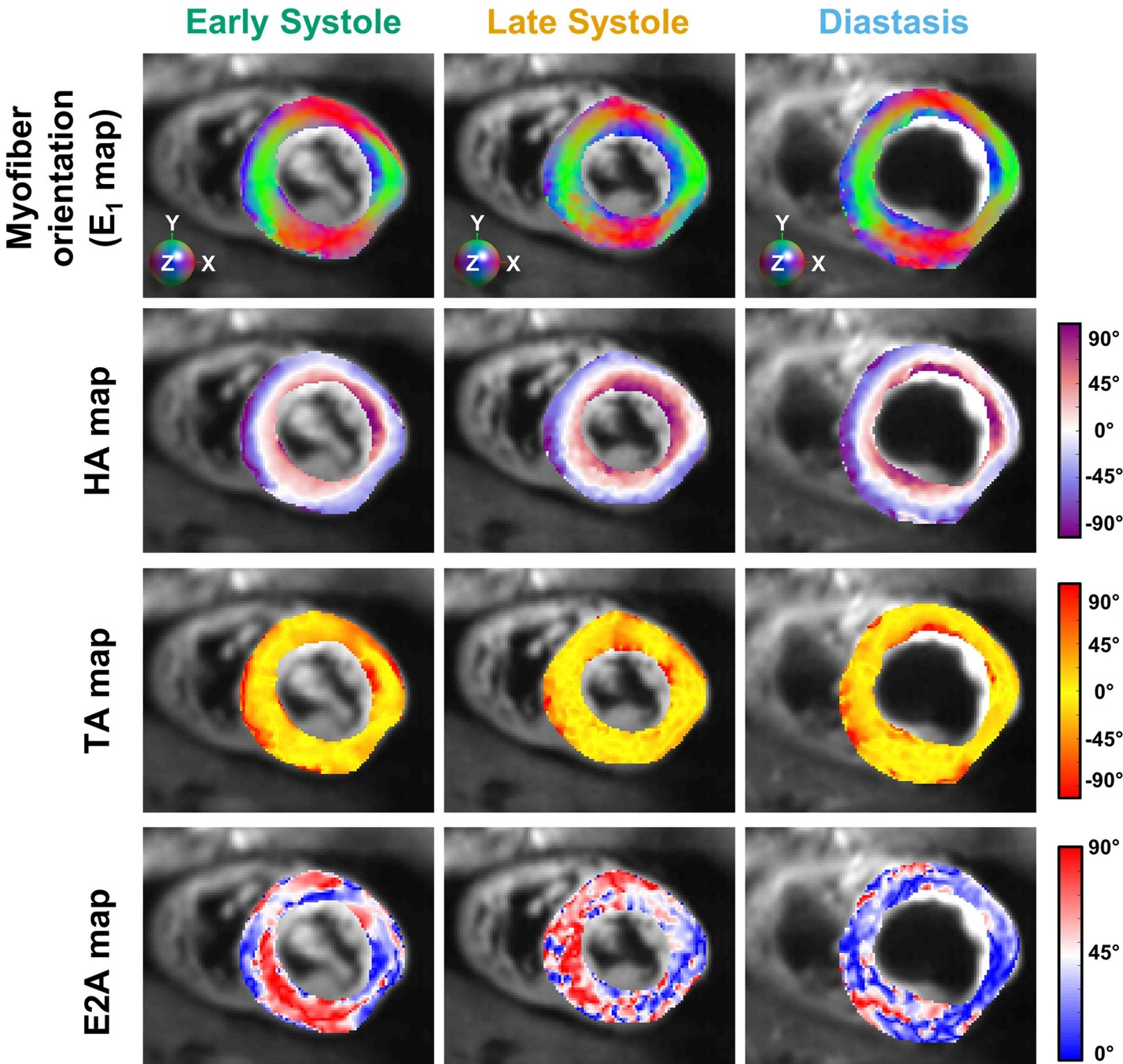

**Fig 4. Reconstructed diffusion tensor data at early systole, late systole, and diastasis for a representative volunteer.** From top to bottom: RGB map of the primary eigenvector ($E_1$) orientation, HA map, TA map, and E2A map (E2A is reported as absolute value).

(45.2 [42.1, 49.0]˚ vs 20.7 [16.6, 26.4]˚). Across cardiac phases, E2A distributions were statistically different (p = 0.001). After post-hoc correction pair-wise statistical differences were only found between late systole and diastasis (p = 0.001), but not between early and late systole (p = 0.065) nor diastasis and early systole (p = 0.576).

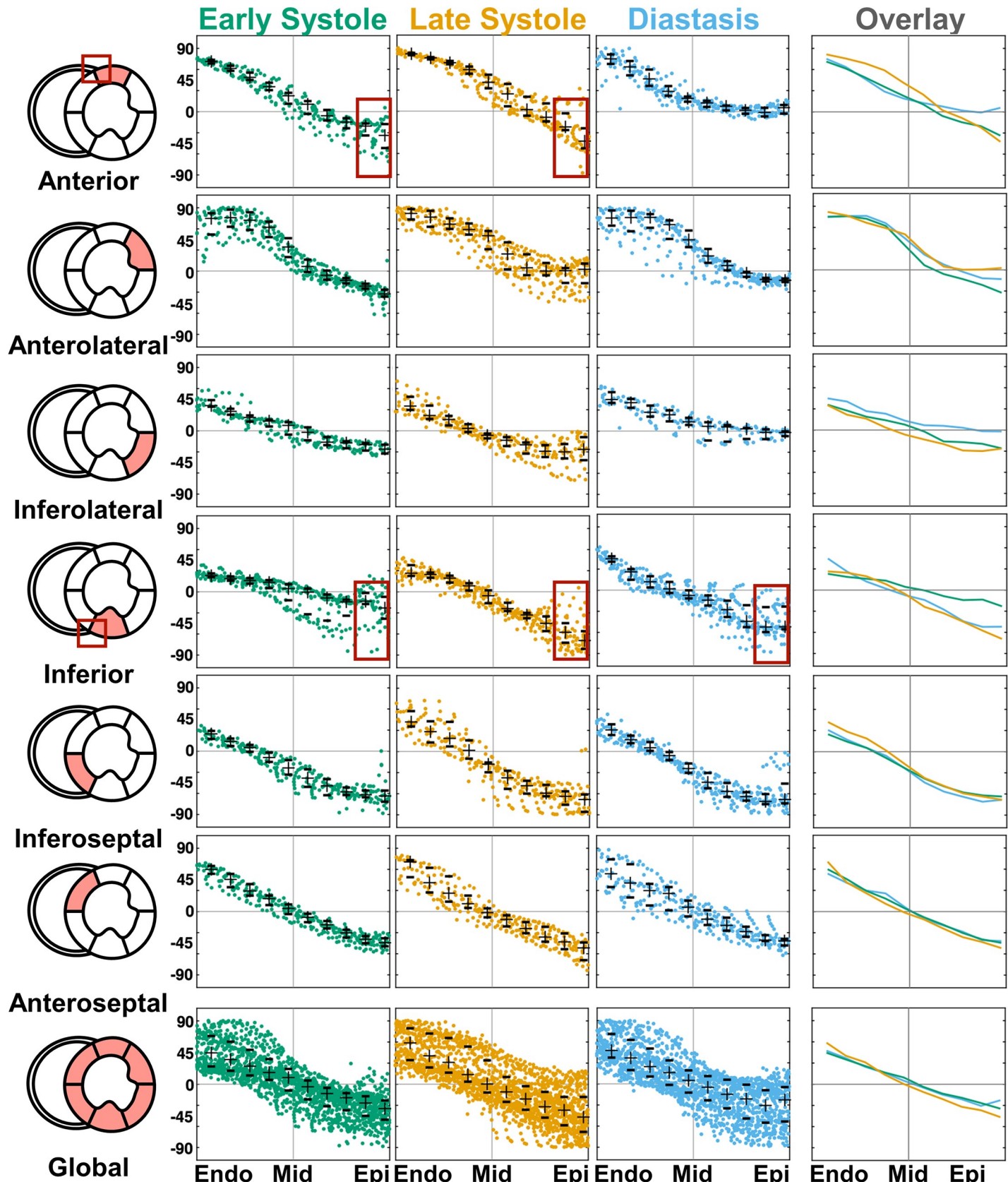

**Fig 5. Transmural HA distributions across six mid-ventricular AHA segments in early systole, late systole, and diastasis in a representative volunteer.** Bottom row shows combined HA distribution across all AHA segments. Rightmost column shows the median profile of the HA distribution colored by cardiac phase. Crosses

and hash marks denote the median and interquartile range of HA at a single transmural depth. Note the different HA distributions across the AHA segments, and the complex HA distribution due to the RV insertion points (red boxes) in the anterior and inferior segments.

## Mean diffusivity, fractional anisotropy, and eigenvalues

Fig 8 shows MD, FA, and the ratio of $\lambda_2$ over $\lambda_1$ as a function of TD across the volunteers for the three cardiac phases. A higher MD was found at early systole compared to late systole (1.79 [1.69, 1.92] vs 1.53 [1.5, 1.62] x $10^{-3}$ mm$^2$/s) and compared to diastasis (1.64 [1.62, 1.69] x $10^{-3}$ mm$^2$/s). Similarly, a higher primary diffusivity $\lambda_1$ (2.52 [2.35, 2.64] vs 2.08 [2.06, 2.15] x $10^{-3}$ mm$^2$/s) and higher secondary diffusivity $\lambda_2$ (1.68 [1.62, 1.85] vs 1.45 [1.42, 1.51] x $10^{-3}$ mm$^2$/s) were observed at early systole vs late systole (S3 Fig). No changes were observed for the ratio $\lambda_{2/1}$ between early systole 0.70 [0.68, 0.72] A.U., late systole 0.70 [0.70, 0.71] A.U., and diastasis 0.69 [0.69, 0.70] A.U. FA was also similar across cardiac phases, 0.36 [0.33, 0.40] A.U. at early systole, 0.33 [0.33, 0.35] A.U. at late systole and 0.35 [0.34, 0.36] A.U. at diastasis. Overall, statistical differences were found across cardiac phases for MD (p = 0.006), $\lambda_1$ (p = 0.002), $\lambda_2$ (p = 0.009) but not for $\lambda_3$ (p = 0.153), $\lambda_{2/1}$ (p = 0.54) or FA (p = 0.32). For MD, $\lambda_1$ and $\lambda_2$, respectively pair-wise comparisons showed significant statistical differences between early and late systole (p = 0.005, p = 0.002, p = 0.009) but not for late systole and diastole (p = 0.293, p = 0.166, p = 0.193) nor diastole and early systole (p = 0.589, p = 0.562, p = 1).

## Reproducibility of helix angle and E2A

Intra- and inter-observer analyses were performed using two observers, each segmenting two sets of LV ROIs. Intra- and inter-observer variations for E2A and HAR are displayed in S4 Fig.

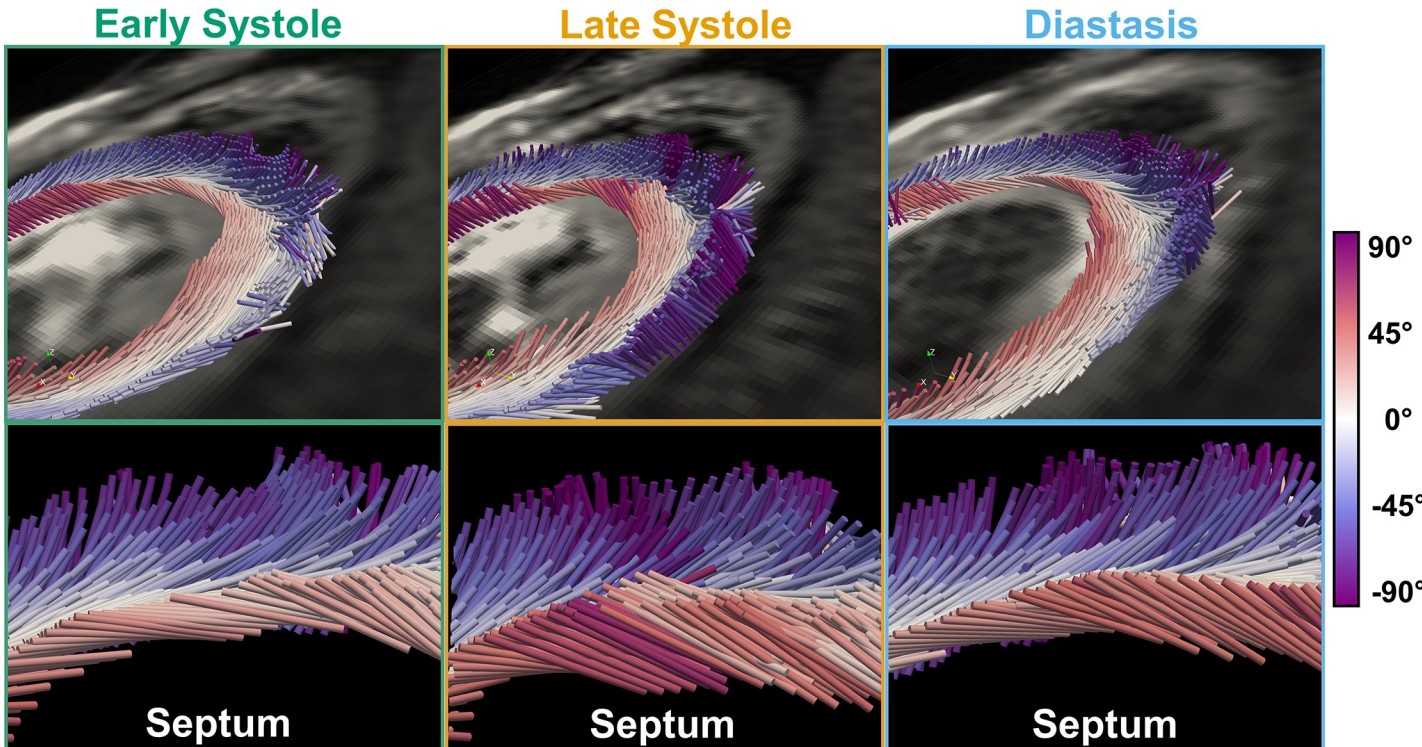

**Fig 6. 3D representation of myofiber orientations at early systole, late systole, and diastasis for a representative volunteer.** (Top) Primary eigenvectors colored by HA. (Bottom) Zoomed-in view of the septal wall. Note the steeper endocardial HA at late systole.

**Table 1. Helix angle and helix angle range per cardiac segments.**

| | HA (°): Anterior Segment | | | |
| --- | --- | --- | --- | --- |
| | Endo | Mid | Epi | HAR |
| Early Sys. (n = 7) | 62.7 [59.8, 64.5] | -1.6 [-5.6, 8.6] | -28.8 [-44.2, -23.6] | 88.6 [85.2, 96.3] |
| Late Sys. (n = 9) | 65.2 [64.6, 68.9] | 13.2 [6.3, 17.4] | -33.9 [-65.3, -23.2] | 111.3 [91.9, 119.0] |
| Diastasis (n = 6) | 60.3 [51.9, 69.9] | 15.1 [9.2, 18.4] | -27.0 [-39.1, -15.5] | 95.2 [71.3, 100.0] |
| | **HA (°): Anterolateral Segment** | | | |
| | Endo | Mid | Epi | HAR |
| Early Sys. (n = 7) | 61.7 [24.8, 76.8] | -7.6 [-11.4, -1.5] | -35.1 [-38.9, -30.0] | 76.6 [63.4, 108.7] |
| Late Sys. (n = 9) | 79.5 [68.7, 79.8] | 25.8 [16.3, 29.3] | -36.5 [-46.3, -14.7] | 105.2 [94.2, 118.9] |
| Diastasis (n = 6) | 65.5 [50.4, 74.6] | 16.9 [13.0, 23.4] | -27.3 [-40.1, -25.2] | 92.2 [82.4, 99.3] |
| | **HA (°): Inferolateral Segment** | | | |
| | Endo | Mid | Epi | HAR |
| Early Sys. (n = 7) | 49.7 [35.5, 59.9] | -9.6 [-12.3, 10.7] | -31.2 [-38.7, -14.5] | 77.7 [65.7, 89.6] |
| Late Sys. (n = 9) | 58.4 [52.2, 62.4] | -2.8 [-5.7, 6.2] | -33.1 [-43.8, -31.8] | 94.8 [84.0, 109.5] |
| Diastasis (n = 6) | 60.5 [56.7, 65.8] | 6.5 [-5.1, 11.4] | -32.5 [-45.4, -23.2] | 96.6 [71.1, 107.1] |
| | **HA (°): Inferior Segment** | | | |
| | Endo | Mid | Epi | HAR |
| Early Sys. (n = 7) | 40.4 [37.0, 45.3] | 12.5 [10.5, 16.4] | -12.0 [-16.7, 2.3] | 47.4 [40.7, 54.0] |
| Late Sys. (n = 9) | 44.9 [36.3, 53.1] | 10.9 [-4.7, 15.4] | -42.1 [-47.7, -18.5] | 87.5 [66.4, 96.0] |
| Diastasis (n = 6) | 40.6 [34.4, 50.3] | -0.9 [-8.8, 9.9] | -26.0 [-44.5, -13.4] | 67.0 [63.3, 82.7] |
| | **HA (°): Inferoseptal Segment** | | | |
| | Endo | Mid | Epi | HAR |
| Early Sys. (n = 7) | 32.4 [30.4, 37.0] | 7.9 [6.7, 12.8] | -50.8 [-58.6, -43.5] | 81.5 [74, 95.6] |
| Late Sys. (n = 9) | 42.9 [36.2, 47.4] | 7.4 [0.3, 8.6] | -48.6 [-61.3, -31.1] | 89.3 [82.2, 99.1] |
| Diastasis (n = 6) | 43 [32.3, 50.0] | -3.8 [-6.4, 1.1] | -64.9 [-69.4, -61.1] | 100.3 [93.1, 114.2] |
| | **HA (°): Anteroseptal Segment** | | | |
| | Endo | Mid | Epi | HAR |
| Early Sys. (n = 7) | 51.5 [40.5, 54.6] | -1.5 [-5.0, 3.7] | -69.6 [-71.2, -40.8] | 106.5 [91.7, 117.2] |
| Late Sys. (n = 9) | 42.6 [38.2, 44.5] | -0.8 [-5.0, 5.7] | -57.9 [-63.3, -45.5] | 95.4 [92.6, 105.4] |
| Diastasis (n = 6) | 53.1 [48.0, 60.4] | -1.1 [-5.0, 7.3] | -48.3 [-50.3, -39.0] | 101.6 [89.3, 103.7] |

HA values per AHA segment given as Median [Q1, Q3] across volunteers in Early systole (Early sys.), Late systole (Late Sys.), and Diastasis. HAR values were *first* calculated per volunteer and then reported as Median [Q1 Q3] across volunteers.

The ICC for HAR was 76 [61, 88] % overall and 84 [73, 91] % for the inter-observer analysis. The intra-observer ICC was 72 [45, 87] % for Observer-1 and 92 [83, 97] % for Observer-2. For E2A values, the ICC was 99% overall, across intra- and inter-observer analyses. Median values for E2A and HAR for the three cardiac phases per segmentation are given in S1 Table. Intra- and inter-observer variations did not affect the overall median nor the trends of E2A and HAR across cardiac phases.

## Discussion

A second order motion compensated diffusion encoding gradient waveform design was used together with a SE approach to acquire cDTI images at a high spatial resolution of 1.6x1.6x8 mm$^3$. To the best of our knowledge, this represents the finest spatial resolution acquired for *in vivo* human cDTI. The TD scout approach used in this work allowed a subject-specific calibration of the trigger delay for cDTI. Using this prospective calibration, two to three cardiac

**Table 2. Delta helix angle range per cardiac segments.**

| Segment | Δ Helix Angle Range (°) Median [Q1, Q3] | | |
|---|---|---|---|
| | From Early systole to Late systole (n = 7) | From Late systole to Diastasis (n = 6) | From Diastasis to Early systole (n = 6) |
| Anterior | 27.5 [6.9, 45.1] | -31.0 [-35.9, -17.6] | 4.6 [-18.6, 26.8] |
| Anterolateral | 14.6 [-21.0, 43.7] | -22.6 [-35.9, -6.0] | -19.7 [-31.2, -9.4] |
| Inferolateral | 27.1 [-2.7, 70.3] | -4.6 [-45.3, 32.7] | -27.4 [-39.4, -15.0] |
| Inferior | 28.5 [15.1, 52.9] | -13.4 [-13.2, -2.5] | -11.5 [-40.4, 22.2] |
| Inferoseptal | 13.5 [3.7, 24.4] | 21.6 [-15.2, 23.7] | -37.4 [-69.5, -7.4] |
| Anteroseptal | -1.1 [-9.6, 17.3] | 9.7 [-13.0, 9.0] | -12.1 [-19.2, -4.1] |
| Global | 21.8 [17.1, 25.4] | -4.2 [-1.3, 3.7] | -11.8 [-29.0, 3.5] |

ΔHAR values per AHA segment given as Median [Q1, Q3] across volunteers. ΔHAR values were _first_ calculated per volunteer according to the available cardiac phases and then reported as Median [Q1 Q3] across volunteers.

phases were identified for _in vivo_ imaging of cardiomyocyte mobility. This approach characterized both global and regional cardiomyocyte orientations and demonstrated significant increases of HAR and E2A during contraction.

## Multi-phase high resolution SE cDTI

The _in vivo_ application of cDTI has been limited due to the sensitivity to cardiac bulk tissue motion, leading to the development of the STEAM sequence [23]. While effective at compensating for cardiac motion and providing a very good diffusion sensitivity, STEAM inherently leads to a significantly extended acquisition time, multiple breath-holds, and limited SNR-efficiency [9], all of which limit the spatial resolution [12]. The SE approach provides inherently higher SNR efficiency and can be used in conjunction with navigator-based free-breathing strategies that enable using a longer TR, thereby leading to more complete longitudinal magnetization recovery between acquisitions [24] for even more SNR. However, diffusion SE acquisitions require a motion compensated ($M_1 = M_2 = 0$) gradient waveform design, which incurs a TE penalty for a fixed b-value. This TE penalty can be partially reduced by optimizing the sequence parameters and the gradient waveform. In this study, the sequence parameters

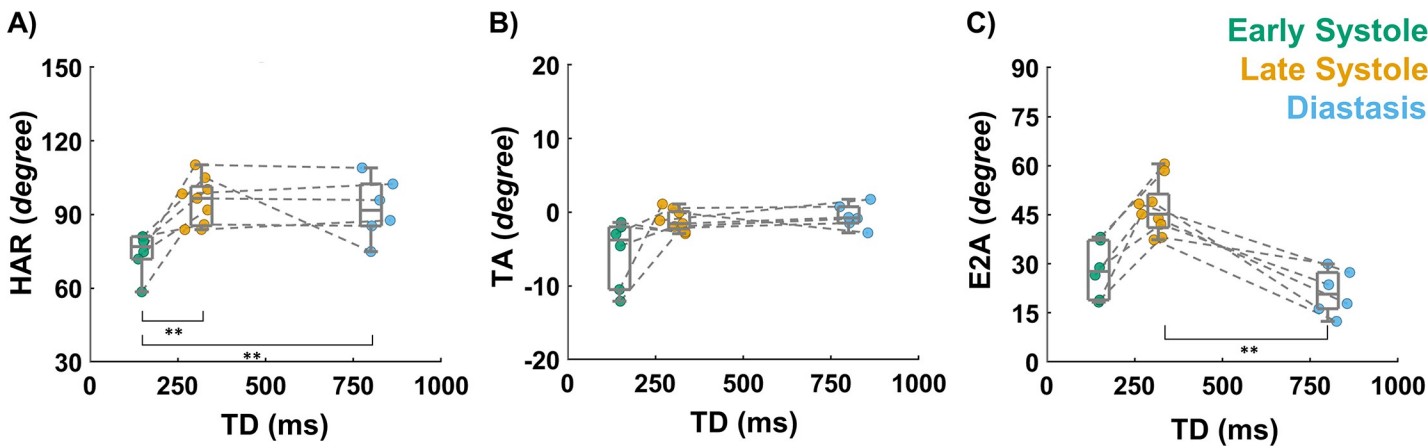

**Fig 7. Angle statistics at early systole, late systole, and diastasis across all volunteers.** Each volunteer corresponds to a circular marker. Median and IQR are reported across volunteers. (A) Helix Angle Range (HAR), (B) Transverse Angle (TA), and (C) E$_2$ Angle (E2A) [** p-value < 0.01].

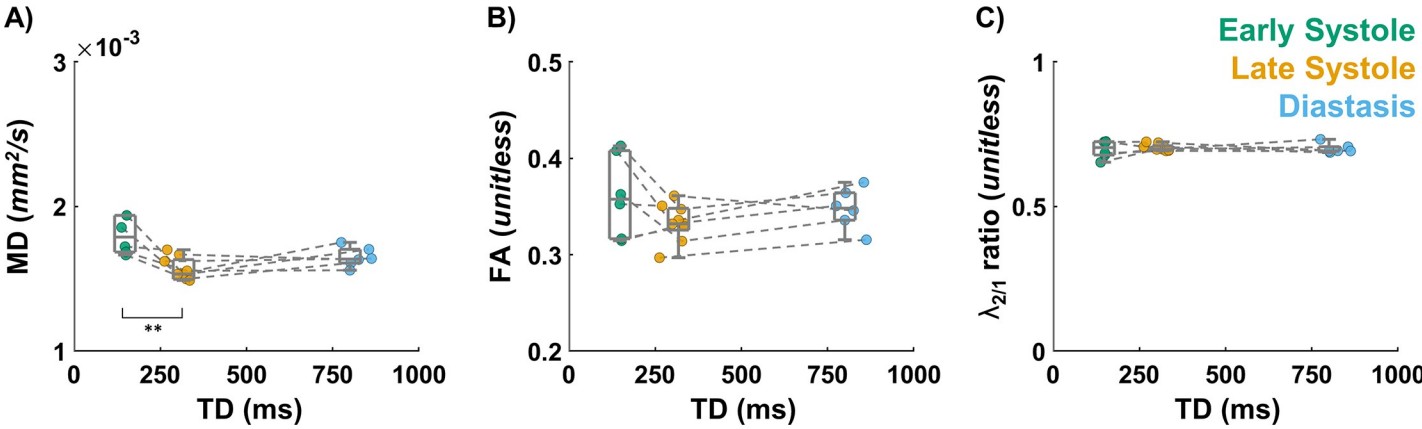

**Fig 8. Tensor invariants statistics at early systole, late systole, and diastasis across all volunteers.** Each volunteer corresponds to a circular marker. Median and IQR are reported across volunteers. (A) Mean diffusivity (MD), (B) Fraction of Anisotropy (FA), and (C) ratio of second and first eigen values ($\lambda_{2/1}$) [** p-value < 0.01].

were manually optimized to minimize the TE contribution from the EPI echo train length, which reduces the overall TE and EPI distortions. The diffusion encoding gradient waveform used a symmetric design to minimize $M_2$ based on the sequence parameters [13]. After this in-line gradient waveform design, a residual dead-time ($\delta_d$, Fig 1C) of only 4ms remained. This dead-time could be further reduced by using a convex optimization approach for asymmetric gradient design [14]. Under these circumstances the benefits of using an asymmetric gradient waveform design would not have afforded significant TE savings.

Gradient waveforms designed with a nulled second order moment are effective and efficient approaches to minimize artifacts induced by intravoxel constant velocity or constant acceleration. However, cardiac deformation is complex and differs from subject to subject. As a result, these linear motion assumptions do not apply to all cardiac phases nor all pixels, which can generate substantial diffusion signal loss in an unpredictable manner. The TD scout calibration was designed to efficiently identify cardiac phases with little signal dropout for each subject. This yields a fast tool for prospectively determining which cardiac phases are best suited for diffusion weighted imaging. As a prospective tool, the TD scout reduces exam time as it ensures that the subsequent cDTI acquisition is performed at the best cardiac phase, thereby avoiding long manual adjustments and scan iteration. Similar approaches to the TD scout have been used by Stoeck et al. [15] and Rapacchi et al. [25] to retrospectively identify acceptable trigger times for cardiac diffusion imaging.

The success rates obtained in this work (78%, 100% and 67% in early, late systole, and diastasis respectively) are close to the ones found by Scott et al [16] using a similar spin echo M1M2 motion compensation as the one used in the current study. The success rate below 100% in early systole and diastasis shows the limitations of this motion compensation approach. As shown in Fig 3, signal dropout due to cardiac motion was observed, particularly in diastole while most of the systolic phases were available for imaging. The cardiac phase imaged in diastole corresponded to diastasis, where the LV motion pauses during filing before atrial systole begins. However, diastasis usually shortens as the heart rate increases and the beat-to-beat variation in cardiac motion may not respect the linear velocity/acceleration assumption behind the M1M2 waveform. Previous works have shown that non-motion compensated waveforms combined with a signal recovery algorithm [25, 26] were able to reliably scan in diastole because of a significantly shorter TE leading to a very small temporal footprint.

Typical in-plane spatial resolutions in clinical cDTI studies range from 2x2 mm$^2$ to 3x3 mm$^2$, usually limited by low SNR. Because HA varies transmurally, higher spatial resolution could play an important role in improving the assessment of regional cardiomyocyte orientation and mobility. Previously McClymont et al. [27] have demonstrated that typical spatial resolutions could underestimate the HAR by up to 18°, in part due to partial volume effects. In this study, high spatial resolution enabled the detailed analysis of *in vivo* HA per AHA segment, revealing distinct microstructural organization that varies per-segment. However high resolution remains challenging, particularly in the inferolateral segment which is prone to strong geometrical distortion due to the lung/heart/liver interface [28].

## Microstructural mobility

In this work, both global and regional HA mobility and a significant increase in HAR during contraction were observed. The HAR in early systole 76.9 [72.6, 80.5]°, late systole 96.6 [85.9, 100.3]°, and in diastasis 91.7 [85.9, 100.8]° agree with early reports of *ex vivo* cDTI animal data. Omann et al. [29], using submillimeter *ex vivo* DTI in swine hearts, reported an LV HAR of 80° and 71° in contracted and relaxed states. These HAR trends are repeatedly observed in other *ex vivo* swine studies, with reported HAR of 109° and 89° [8], and 107° and 87° [30] in the contracted and relaxed states, respectively. In rat hearts, Chen et al. [31] reported a HAR of 111°, 134°, and 105° in early systole, peak systole, and diastole, respectively. *In vivo* cDTI studies of the healthy human LV have also shown a steepening of HA during systole. Stoeck *et al.* [22] reported a HAR of 56° and 72° in diastole and systole, while von Deuster *et al.* [32] reported a HAR of 54° and 77°.

A detailed characterization of healthy HA mobility is valuable in providing a baseline to study the microstructural mechanisms of LV dysfunction. Indeed, several studies have implicated deleterious HA remodeling and abnormal mobility in heart disease. For example, an increase in the transmural slope of cardiomyocyte elevation was recently described in infarcted tissue [33] with respect to remote myocardium. von Deuster *et al.* [32] observed a decrease in HA mobility from diastole to systole in patients with dilated cardiomyopathy. Recently, Gotschy *et al.* [11] demonstrated the link between global longitudinal strain and HA mobility in patients with cardiac amyloidosis. Clinical studies on hypertrophic cardiomyopathy [3] appear to show a steepening of HA during contraction, although the reporting primarily focused on E2A changes.

We also observed an increase of E2A in late systole compared to early systole and diastasis. As shown in Fig 2, E2A represents a combination of projections involving both $E_1$ and $E_2$, and is usually attributed to the cardiomyocyte sheetlet direction [3]. In this work, E2A increased from 27.7 [20.8, 35.1]° in early systole to 45.2 [42.1, 49]° in late systole, and decreased to 20.7 [16.6, 26.4]° in diastasis. Using a STEAM-based approach and at slightly different cardiac phases, studies in healthy controls demonstrate E2A changing from 65° to 18° [8] and from 56° to 24° [3] from peak systole to diastole, respectively. In the same studies and in patients with hypertrophic cardiomyopathy, E2A was reported to decrease from 74° to 48° [8] and from 64° to 47° [3] from peak systole to diastole, respectively. E2A changes were also observed in patients with dilated cardiomyopathies [8] and amyloidosis [11], showing the importance of measuring E2A in addition to HA as a diagnostic biomarker and to better understand how changes in E2A relate to ventricular contraction.

Although HAR and E2A mobility trends are consistent across cDTI studies, there is a broad range in the reported magnitude of HAR and E2A changes during contraction and filling. As shown by Scott et al [16], these variations could be attributed in part to difference in sensitivity (diffusion time) between SE and STEAM imaging. In addition, a collection of artifacts

(motion, off-resonance, chemical shift of fat, etc.) are also present in cDTI, with a different effect in SE and STEAM imaging [9, 16, 22, 34]. Finally, the angular metrics used to represent the microstructure (namely HA, TA, E2A, HAR) are based on circumferential and radial directions constructed from manually defined segmentations. This inter-observer dependence is particularly significant in evaluating the HAR metric based on the HA in proximity of the endocardial and epicardial surfaces. This can be seen in the intra-and inter-observer study presented in S4 Fig, where we report more variation across subjects for HAR than E2A. However as shown in S1 Table, median trends during the cardiac cycle remain consistent for both intra and inter-observer analyses.

Cardiomyocyte mobility has been reported in a wide range of cardiac phases across cDTI literature. As shown in the TD scout measures SE approaches can measure most of contraction up to late systole (~150 to 300ms), and during the diastasis phase in diastole. In this work, the complete cardiomyocyte mobility cycle was not observed, as the cardiomyocytes in diastasis do not completely return to their initial configurations in early systole. The return in cardiomyocyte orientations to complete the cardiac cycle must occur outside the window of the cardiac phases captured in this work. Future studies are needed to determine precisely when HA and E2A return to their configuration at the beginning of systole.

### Diffusivity values at early systole, late systole, and diastasis

The MD and FA values reported in this work are in the range of previously reported values in healthy volunteers using spin-echo sequences. Statistically higher MD values were found at early systole 1.79 [1.69, 1.92] x $10^{-3}$ mm$^2$/s compared to late systole 1.53 [1.5, 1.62] x $10^{-3}$ mm$^2$/s or diastasis 1.64 [1.62, 1.69] x $10^{-3}$ mm$^2$/s. In a previous work by Stoeck et al. [22], a higher MD was found in systole 1.2 ± 0.2 x $10^{-3}$ mm$^2$/s compared to diastole 0.9 ± 0.1 x $10^{-3}$ mm$^2$/s using a STEAM approach. Scott et al. [16].found the opposite trend with an MD of 1.02 [0.14] x $10^{-3}$ mm$^2$/s in systole, 1.07 [0.12] x $10^{-3}$ mm$^2$/s at the systolic sweet spot and 1.13 [0.08] x $10^{-3}$ mm$^2$/s in diastole using a STEAM approach. In the same study using a SE sequence, MD was 1.46 [0.43] x $10^{-3}$ mm$^2$/s in systole, 1.70 [0.18] x $10^{-3}$ mm$^2$/s at the systolic sweet spot and 1.78 [0.34] x $10^{-3}$ mm$^2$/s in diastole. In our previous work [35] using SE, a higher MD was found in diastole 1.91 ± 0.34 x $10^{-3}$ mm$^2$/s compared to systole 1.58 ± 0.09 x $10^{-3}$ mm$^2$/s. Values were reported as mean ± standard deviation, Median [Interquartile Range] or Median [Quartile 1, Quartile 3]. The source of this increase could be explained by several effects. The higher MD in early systole could be partially due to micro-perfusion, which is known to increase during filling and whose signal could not be fully spoiled by the motion compensated diffusion encoding gradients [36]. Despite our use of a motion compensated waveform, unwanted cardiac bulk motion present in early systole could artificially increase MD [15]. However, in this work, the diffusivity differences are unlikely due to a change in cellular/extra-cellular compartments between cardiac phases. The b-value (350 s/mm$^2$) and the diffusion encoding time used here ($\Delta$ = 40ms) were too small to sensitively capture these compartmental changes [16], which is also confirmed by fact that the ratio $\lambda_{2/1}$ remains constant across cardiac phases.

### Limitations

A first limitation of this study is the single-slice heart coverage acquired with a long TR to maximize SNR. This approach limits characterizing cardiomyocyte mobility throughout the heart and limits overall SNR efficiency. Several approaches have been proposed to improve the spatial coverage without increasing scan time, such as simultaneous multi-slice acquisition [37] and the slice-following navigator technique [24]. Both temporal and spatial coverage could be

improved using advanced dual cardiac phase encoding [22]. All these approaches would improve SNR efficiency and could offer improved clinical applicability of the presented protocol.

Second, all three cardiac phases of the cDTI datasets were acquired at the same spatial location for each volunteer. However, the myocardium passes through the imaging plane as the heart contracts and fills and thus the tissue imaged in the three cardiac phases may differ slightly. The mid-ventricular longitudinal motion from base to apex was approximately 4mm from early to late systole, 4mm from diastasis to early systole and -8mm between late systole and diastasis (estimated using DENSE imaging on a subset of the study subjects). Considering the slice thickness (8mm) and the relative longitudinal homogeneity of the cardiomyocyte orientation [2], it is possible that through-plane tissue motion mildly influences the microstructural mobility reported in this cDTI study. Few multi-cardiac phase *in vivo* cDTI studies have considered the longitudinal displacement of the heart by estimating this displacement prospectively [3].

## Conclusion

We demonstrate that it is possible to acquire high-resolution cDTI datasets in ~7 minutes per cardiac phase during free-breathing. This is compatible with clinical time constraints and suitable for patients with respiratory problems. In healthy volunteers, *in vivo* regional HA mobility was reported at three distinct cardiac phases and suggests that on average cardiomyocytes reorient more longitudinally during contraction, while sheetlets tilt in the direction of LV wall thickening. In the future, these observations may provide clinical insight into LV function in healthy and diseased hearts. Given the intricate structural and microstructural phenotypes associated with heart disease [4, 5, 38], high resolution *in vivo* cDTI has the potential to offer a better understanding of the complex link between microstructural remodeling and the underlying cardiac dysfunction at a patient-specific level.

## Supporting information

**S1 Fig. Automatic estimation of the signal per cardiac phase imaged during the TD scout.** A) Three images corresponding to three diffusion encoding directions x, y, and z are acquired during free breathing for each cardiac phase. B) The three diffusion weighted images are averaged together to generate a trace image. C) The trace image is then circularly cropped with a 48mm radius to only include the central zone of the image corresponding to the heart. D) The mean signal and standard deviation (SD) are computed using the cropped trace image for each cardiac phase.
(TIF)

**S2 Fig. Example of the filtering method adopted to remove the projection error in computing the Helix Angle (HA).** A) The unprocessed distribution of HA as a function of wall depth. B) The HA distribution is fitted to a simple linear model (*ax+b*) and the data points outside one standard deviation above and below the linear model are classified as outliers. C) Finally, the outliers which are negative from Endo to Mid or positive from Mid to Epi are shifted by 180˚ in order to flip them in the correct segment. Yellow arrows show the main regions in the cDTI image affected by these outliers.
(TIF)

**S3 Fig. Diffusivity statistics at early systole, late systole, and diastasis across all volunteers.** Each volunteer corresponds to a circular marker. Median and IQR are reported across volunteers. (A) Primary diffusivity ($\lambda_1$), (B) secondary diffusivity ($\lambda_2$), and (C) tertiary diffusivity

($\lambda_3$). [** p-value $< 0.01$]
(TIF)

**S4 Fig. Intra-and inter-observer variability in Helix Angle Range (HAR) (top) and $E_2$ Angle (E2A) (bottom) computed across two segmentations (intra-observer) and two observers (inter-observer).** The intraclass correlation coefficient (ICC) for HAR was 76% overall and 84% inter-observer. The intra-observer ICC was 72% for Observer-1 and 92% for Observer-2. The ICC for E2A was 99% overall, inter and intra-observer.
(TIF)

**S1 Table. Median Helix Angle Range (HAR) and E2 Angle (E2A) computed by two observers across volunteers.** Each observer segmented the data twice. Intra-and inter-observer variability does not affect the overall changes in HAR and E2A across the analyzed cardiac phases.
(TIF)

## Author Contributions

**Conceptualization:** Kévin Moulin, Ilya A. Verzhbinsky, Luigi E. Perotti, Daniel B. Ennis.

**Data curation:** Kévin Moulin, Ilya A. Verzhbinsky, Nyasha G. Maforo, Luigi E. Perotti.

**Formal analysis:** Kévin Moulin, Ilya A. Verzhbinsky, Luigi E. Perotti.

**Funding acquisition:** Kévin Moulin, Ilya A. Verzhbinsky, Daniel B. Ennis.

**Investigation:** Kévin Moulin, Ilya A. Verzhbinsky, Luigi E. Perotti.

**Methodology:** Kévin Moulin, Ilya A. Verzhbinsky, Luigi E. Perotti.

**Project administration:** Kévin Moulin, Daniel B. Ennis.

**Resources:** Kévin Moulin, Daniel B. Ennis.

**Software:** Kévin Moulin.

**Supervision:** Kévin Moulin, Luigi E. Perotti, Daniel B. Ennis.

**Validation:** Kévin Moulin, Luigi E. Perotti, Daniel B. Ennis.

**Visualization:** Kévin Moulin.

**Writing – original draft:** Kévin Moulin, Ilya A. Verzhbinsky, Nyasha G. Maforo, Luigi E. Perotti, Daniel B. Ennis.

**Writing – review & editing:** Kévin Moulin, Ilya A. Verzhbinsky, Nyasha G. Maforo, Luigi E. Perotti, Daniel B. Ennis.

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
