## [Decision Letter · Decision Letter 0]

19 Jun 2020

PONE-D-20-15447

Probing Cardiomyocyte Mobility with Multi-Phase Cardiac Diffusion Tensor MRI

PLOS ONE

Dear Dr. Moulin,

Thank you for submitting your manuscript to PLOS ONE. After careful consideration, we feel that it has merit but does not fully meet PLOS ONE’s publication criteria as it currently stands. Therefore, we invite you to submit a revised version of the manuscript that addresses the points raised during the review process.

 All issues raised by expert Reviewers are required.

We look forward to receiving your revised manuscript.

Kind regards,

Vincenzo Lionetti, M.D., PhD

Academic Editor

PLOS ONE

Journal Requirements:

2. Thank you for including your ethics statement:  "Nine healthy volunteers (N=9) were recruited for an MRI exam after obtaining signed statements of informed consent under an IRB approved and HIPPA compliant protocol".   

"This work was supported by NIH/NHLBI R01-HL131975, R01-HL131823, and K25-HL135408 grants, and by NSF DGE 1650604 grant. The content of this manuscript issolely the responsibility of the authors and does not necessarily represent the official viewsof the National Institutes of Health."

Reviewers' comments:

Reviewer's Responses to Questions

**Comments to the Author**

1. Is the manuscript technically sound, and do the data support the conclusions?

Reviewer #1: Yes

Reviewer #2: Yes

2. Has the statistical analysis been performed appropriately and rigorously? 

Reviewer #1: Yes

Reviewer #2: No

3. Have the authors made all data underlying the findings in their manuscript fully available?

Reviewer #1: Yes

Reviewer #2: No

4. Is the manuscript presented in an intelligible fashion and written in standard English?

Reviewer #1: Yes

Reviewer #2: Yes

5. Review Comments to the Author

Reviewer #1: The Authors present the results of a novel spin-echo sequence to acquire cardiac diffusion tensor imaging (cDTI) with high spatial resolution; three distinct cardiac phases have been chosen using a TD - scout sequence. Overall, the article is interesting and well written.

-A major limitation (as acknowledged by the Authors) is the through-plane motion between different cardiac phases ("approximately 6.5 mm for a longitudinal strain of -0.15"). Was this longitudinal motion calculated on the long axis cine images? In the discussion, I would suggest the Authors to comment on the possibility of a three-dimensional acquisition instead of a single 2D midventricular short-axis slice, to avoid this limitation.

-Moreover, in the discussion I would suggest the Authors to comment on the possibility of acquiring 4 or more TDs, at least under certain conditions (lower heart rate, highly compliant patients...)

-In the introduction (first paragraph), after explaining the HA (E1), the Authors should also briefly define the sheetlet angle (SA, or more precisely referred to as the secondary eigenvector E2 angle); indeed, both parameters are analyzed in the manscript, as depicted in Figure 2. Also the TA (transverse angle) might be cited here.

-Was there a relationship between image quality and heart rate?

-Did the Authors perform intra-observer and inter-observer reproducibility of post-processing analysis?

-Did the Authors perform test-retest (inter-study) reproducibility?

Reviewer #2: This manuscript describes the use of a motion-compensated spin-echo sequence to acquire relatively high resolution cardiac diffusion tensor imaging (cDTI) data in 9 healthy subjects at multiple cardiac phases. The cDTI data provides measures of cardiac microstructure. Images acquired in early systole, late systole and diastasis were successful in 67%, 100% and 77% of cases respectively. The authors provide helix angle range, sheetlet angle and tensor invariants in multiple cardiac phases. The stated objective of the work was to characterize cardiomyocyte orientations and mobility throughout the cardiac cycle at high spatial resolution and this was achieved.

The study is novel in acquiring in-vivo cDTI data in humans at higher in-plane spatial resolution than previous studies (1.6x1.6mm2), although slice thickness is not reduced relative to comparable studies. The optimal trigger delay for each cardiac phase is determined from a separately acquired novel TD scout. This TD scout allows the user to identify the trigger delays with minimal motion related signal loss from a series of images acquired at a range of TD values. Previous studies by other groups have acquired spin-echo cDTI data at similar cardiac phases without the use of a TD scout, albeit at lower in-plane spatial resolution (Scott et al. JCMR 2018, https://doi.org/10.1186/s12968-017-0425-8). While the TD scout method would appear to be a sensible approach to determining the optimal trigger delay, the success rate at each cardiac phase is not substantially improved over Scott 2018 and the authors do not provide a comparison of the success rate using the TD scout vs. a more standard approach (visual assessment based on cine data). Similarly, the high spatial resolution would seem to be a substantial benefit (if the associated SNR loss is not a problem) when assessing differences in helix angle transmurally and between cardiac phases but no comparison is performed with more typical resolutions to show the benefits.

The authors have included links to a code repository containing the post-processing code, which is commendable. They have also provided a comprehensive list of sequence design parameters, which is excellent.

There are minor typos in the work (included in the comments below), but it is otherwise presented well.

The statistical analysis does not appear entirely appropriate, but should be easily addressed and the PLOS requirements for data sharing do not appear to be satisfied (see specific comments below).

Specific comments:

1. Abstract, results – the success rates for early and late systole appear to be switched around.

2. Methods – The text suggests that the EPI readout was 14ms long, while the figure legend (figure 1) says that TEPI was 5ms. It also appears as if TEPI and TEPI-echo are switched around in the legend, but even if so the numbers don’t match the main text.

3. Methods Post-processing – The TD scout images were processed to provide an average image and mean signal. Please clarify what was used to determine the optimal cardiac phases when scanning, was it the quantitative signal intensity or visual appearance?

4. Methods Quantitative analysis – There is much focus in the manuscript on helix angle range (HAR). However, I expect that HAR is highly dependent on the definition of the epi and endocardial borders drawn and therefore could be rather observer dependent. I suggest that additional analysis to provide intra and inter observer variability of the HAR measures used should be performed.

5. Methods – statistical analysis: The data are presented as medians and interquartile ranges, but the statistical comparisons are parametric. The use of median and interquartile range, implies non-parametric data. On what basis were parametric tests used? If the use of parametric tests can be justified then data should be presented as mean and standard deviation. Furthermore, it is usual to perform an ANOVA type test to compare more than two groups and then perform subsequent paired comparisons where differences are identified on ANOVA.

6. Methods – last sentence, typo “Holm-Bonferroni and correction”.

7. Results – It’s unclear where statistical comparisons have been performed and differences found to be non-significant. I suggest quoting p-values for all comparisons performed, whether the result is significant or not.

8. Results – Differences in helix angle and helix angle range are presented for 6 AHA segments. Some of these differences could be the consequence of off-resonance effects rather than changes in microstructure. These off-resonance effects are generally most severe in the inferolateral wall of the LV and least severe in the septum.

9. Discussion – The spin-echo sequence is theoretically much more SNR efficient than the STEAM sequence, but the calculations are typically based on a TR of 1 or 2 cardiac cycles. This study uses a TR=4 RR intervals which is very long and results in a lot of dead time. Can this be improved in future studies by interleaving slices? Presumably in most end expiratory periods there are two cardiac cycles, which could be used to acquire 2 slices with no loss of SNR if angled refocussing pulses are used (as in Stoeck 2014 https://doi.org/10.1371/journal.pone.0107159)

10. Discussion – median missing from E2A values quoted in late systole.

11. Discussion – The variation in E2A and HA between studies is attributed to spatial resolution, which is rather speculative and doesn’t account for other potential contributing factors such as the diffusion time, sensitivity to off-resonance etc.

12. Discussion – typo “were observe[d] among different cardiac phases”.

13. Discussion – The discussion of the reasons behind the changes in tensor invariants between cardiac phases is slightly confusing. If the M1-M2 compensated gradient waveforms are insensitive to blood flow then wouldn’t the signal in both the reference (b=0) and diffusion encoded images (b=350) be equally affected by blood flow? This would make the diffusion tensor invariant to changes in blood flow.

It’s also not clear what the authors think causes the changes in parameters if it is not caused by changes in the microscopic compartments. Presumably the sequence is somewhat sensitive to changes in compartment dimensions otherwise the signal would be isotropic?

14. Discussion, final paragraph – Several studies including Ferreira et al. JCMR 2014 https://doi.org/10.1186/s12968-014-0087-8 have shifted the imaging plane between cardiac phases to account for this potential issue.

15. Figure 3 – this seems to be identical to supplementary figure 3 apart from the normalisation of the x axis in the supplementary figure. The normalised axis seems more logical in my opinion and the inclusion of both seems excessive.

16. Data availability – PLOS one requires that a minimal data set is shared (https://journals.plos.org/plosone/s/data-availability). It does not appear to me that this is the case for this manuscript.

“For example, authors should submit the following data:

- The points extracted from images for analysis.”

6. PLOS authors have the option to publish the peer review history of their article (what does this mean?). If published, this will include your full peer review and any attached files.

Reviewer #1: No

Reviewer #2: No

---

## [Author Response · Author response to Decision Letter 0]

5 Aug 2020

We appreciate the editor and reviewers’ comments and are grateful that they are experts regarding the subject matter. We have worked to address all concerns with the revisions in our resubmitted manuscript. The specific comments from the reviewers has been "Answer to reviewer" document. 

Regarding the editorial comment:

1. We followed PLOS ONE's style requirements

2. We amended the “Ethics Statement” field of the submission form and the manuscript accordingly. 

3. We removed the funding statement from the Acknowledgments Section of the manuscript.

---

## [Decision Letter · Decision Letter 1]

25 Aug 2020

PONE-D-20-15447R1

Probing Cardiomyocyte Mobility with Multi-Phase Cardiac Diffusion Tensor MRI

PLOS ONE

Dear Dr. Moulin,

Thank you for submitting your manuscript to PLOS ONE. After careful consideration, we feel that it has merit but does not fully meet PLOS ONE’s publication criteria as it currently stands. Therefore, we invite you to submit a revised version of the manuscript that addresses the points raised during the review process.

ACADEMIC EDITOR: All issues raised by expert reviewer are required in order to better support new data. Moreover, editing of english grammar and style should be performed.

We look forward to receiving your revised manuscript.

Kind regards,

Vincenzo Lionetti, M.D., PhD

Academic Editor

PLOS ONE

Reviewers' comments:

Reviewer's Responses to Questions

**Comments to the Author**

1. If the authors have adequately addressed your comments raised in a previous round of review and you feel that this manuscript is now acceptable for publication, you may indicate that here to bypass the “Comments to the Author” section, enter your conflict of interest statement in the “Confidential to Editor” section, and submit your "Accept" recommendation.

Reviewer #1: All comments have been addressed

Reviewer #2: (No Response)

2. Is the manuscript technically sound, and do the data support the conclusions?

Reviewer #1: Yes

Reviewer #2: Yes

3. Has the statistical analysis been performed appropriately and rigorously? 

Reviewer #1: Yes

Reviewer #2: Yes

4. Have the authors made all data underlying the findings in their manuscript fully available?

Reviewer #1: Yes

Reviewer #2: Yes

5. Is the manuscript presented in an intelligible fashion and written in standard English?

Reviewer #1: Yes

Reviewer #2: No

6. Review Comments to the Author

Reviewer #1: (No Response)

Reviewer #2: Thank you for revising the manuscript based on my comments and those of the other reviewer. I appreciate the time and effort that you have gone to in addressing my concerns. I feel that the added intra and inter-observer analysis of HAR and E2A is a really good addition to the manuscript. I also appreciate the provision of the associated data and analysis code.

I have a few additional comments mostly related to the additional results added to the paper that I hope can be addressed.

1. The intra and inter observer reproducibility study is interesting, but should be described in the methods and results rather than being first mentioned in the discussion. The same principle applies to the DENSE study and analysis, which should be described in the methods and results.

2. Based on figure 3 and line 239, it appears that the early systolic timepoint was the same trigger delay index (nth TD) for all acquisitions. In which case the TD scout serves more as a method to determine whether cDTI will be possible in a particular phase than to select the correct phase? In that case, the TD scout serves to save time when it demonstrates that cDTI data cannot be acquired at a particular trigger delay, which could be highlighted in the discussion.

3. Similarly line 410 states that diastasis was the most common time available for imaging during diastasis. However, the methods section suggests that the trigger search window for diastole was limited to diastasis (line 170), making this statement in the discussion redundant.

4. Line 353, FA was slightly higher in early systole, but not significantly. In the prior sentence there is a similar magnitude in change in the eigenvalue ratio, which you say is not a change. I would probably avoid suggesting that there is any change in FA for consistency.

5. Line 445: Results in ex-vivo fetal hearts are irrelevant when discussing HA mobility as the ex-vivo data is static.

6. Figure S1. There is a substantial artefact in these example images, which should be acknowledged and the source of which should be included in the legend. Was this artefact also present in the corresponding cDTI images?

7. Supplementary figure S4. I appreciate that the vertical scale of the E2A Bland-Altmann plots is chosen to be similar to that of the HAR, but the range of potential E2A values is much smaller and I would rather be able to see the distribution of E2A differences than maintain the consistent scale between angular measures. Please consider re-plotting with a different vertical scale.

Minor comments and typos. This list may not be complete.

8. line 59, additional full stop.

9. line 76, redundant "imaging" after cDTI.

10. Line 188, inconsistent capitalisation of Eigenvectors (see next line). Also present elsewhere for other terminology.

11. Line 315, additional comma after 100.3] ˚

12. Throughout results, "paired-wised" should read "pair-wise"

13. Line 320, "between early [and] late"

14. line 334: difference[s]

15. Line 446: Gotschy spelling mistake.

16. Line 468: metric[s]

17. Line 475: "across intra and inter trials" doesn't read well.

18. Line 498: intra quartile range -> interquartile range

19. Line 510: Consider removing "However"

20. Line 520: Define the positive direction used for DENSE displacements.

7. PLOS authors have the option to publish the peer review history of their article (what does this mean?). If published, this will include your full peer review and any attached files.

Reviewer #1: No

Reviewer #2: No

---

## [Author Response · Author response to Decision Letter 1]

9 Oct 2020

We appreciate the editor and reviewers’ second round of comments and we have addressed all new concerns with the revisions in our resubmitted manuscript. Our answers to each of the reviewers’ specific comments are in the document "Response to Reviewers" joint to the submission.

---

## [Decision Letter · Decision Letter 2]

26 Oct 2020

Probing Cardiomyocyte Mobility with Multi-Phase Cardiac Diffusion Tensor MRI

PONE-D-20-15447R2

Dear Dr. Moulin,

We’re pleased to inform you that your manuscript has been judged scientifically suitable for publication and will be formally accepted for publication once it meets all outstanding technical requirements.

Kind regards,

Vincenzo Lionetti, M.D., PhD

Academic Editor

PLOS ONE

Additional Editor Comments (optional):

Reviewers' comments:

Reviewer's Responses to Questions

**Comments to the Author**

1. If the authors have adequately addressed your comments raised in a previous round of review and you feel that this manuscript is now acceptable for publication, you may indicate that here to bypass the “Comments to the Author” section, enter your conflict of interest statement in the “Confidential to Editor” section, and submit your "Accept" recommendation.

Reviewer #1: All comments have been addressed

Reviewer #2: All comments have been addressed

2. Is the manuscript technically sound, and do the data support the conclusions?

Reviewer #1: Yes

Reviewer #2: (No Response)

3. Has the statistical analysis been performed appropriately and rigorously? 

Reviewer #1: Yes

Reviewer #2: (No Response)

4. Have the authors made all data underlying the findings in their manuscript fully available?

Reviewer #1: Yes

Reviewer #2: (No Response)

5. Is the manuscript presented in an intelligible fashion and written in standard English?

Reviewer #1: Yes

Reviewer #2: (No Response)

6. Review Comments to the Author

Reviewer #1: (No Response)

Reviewer #2: (No Response)

7. PLOS authors have the option to publish the peer review history of their article (what does this mean?). If published, this will include your full peer review and any attached files.

Reviewer #1: No

Reviewer #2: No

---

## [Editor Report · Acceptance letter]

4 Nov 2020

PONE-D-20-15447R2 

Probing Cardiomyocyte Mobility with Multi-Phase Cardiac Diffusion Tensor MRI 

Dear Dr. Moulin:

I'm pleased to inform you that your manuscript has been deemed suitable for publication in PLOS ONE. Congratulations! Your manuscript is now with our production department. 

Kind regards, 

on behalf of

Prof. Vincenzo Lionetti 

Academic Editor

PLOS ONE